# Paired Comparison of Routine Molecular Screening of Patient Samples with Advanced Non-Small Cell Lung Cancer in Circulating Cell-Free DNA Using Three Targeted Assays

**DOI:** 10.3390/cancers15051574

**Published:** 2023-03-03

**Authors:** David Barthelemy, Gaelle Lescuyer, Florence Geiguer, Emmanuel Grolleau, Arnaud Gauthier, Julie Balandier, Margaux Raffin, Claire Bardel, Bruno Bouyssounouse, Claire Rodriguez-Lafrasse, Sébastien Couraud, Anne-Sophie Wozny, Léa Payen

**Affiliations:** 1Institut of Pharmaceutical and Biological Sciences (ISPB), Claude Bernard Lyon I, 69373 Lyon, France; 2Department of Biochemistry and Molecular Biology, Lyon-Sud Hospital, Hospices Civils de Lyon, 69495 Pierre-Bénite, France; 3Center for Innovation in Cancerology of Lyon (CICLY) EA 3738, Faculty of Medicine and Maieutic Lyon Sud, Claude Bernard University Lyon I, 69921 Oullins, France; 4Circulating Cancer (CIRCAN) Program, Hospices Civils de Lyon, Cancer Institute, 69495 Pierre-Bénite, France; 5Faculty of Medicine and Maieutic Lyon Sud, Claude Bernard University Lyon 1, 69921 Oullins, France; 6Acute Respiratory Disease and Thoracic Oncology, Pneumology Department of Lyon Sud Hospital, Hospices Civils de Lyon, 69495 Pierre-Bénite, France; 7Cellular and Molecular Radiobiology Laboratory UMR CNRS5822/IP2I, Faculty of Medicine and Maieutic Lyon Sud, Claude Bernard University Lyon I, 69921 Oullins, France; 8Department of Bioinformatics, Hospices Civils de Lyon, 69008 Lyon, France; 9INOVOTION, 38700 La Tronche, France

**Keywords:** cfDNA, MRD, early stages, lung cancer, NSCLC, digital PCR, custom validated NGS assay, comparison of methods, performance

## Abstract

**Simple Summary:**

Circulating tumor DNA (ctDNA) samples reflect the total tumor burden and allow longitudinal monitoring of mutational sensitizing alterations in routine use for advanced non-small cell lung cancer (NSCLC). Various assays are developed at high sensitivity and specificity. To drive the choice of the best assay at diagnosis, we compared the clinical performance of an ultra-sensitive Plasma-SeqSensei™ SOLID CANCER IVD kit with the Plasma OncoBEAM^TM^ EGFR V2 assay, or with our custom validated NGS routine assay. Global clinical concordances rates of 75% and 68% were found between the Plasma-SeqSensei™ SOLID CANCER IVD assay with the Plasma OncoBEAM^TM^ EGFR V2 assay, and our custom validated NGS assays. The Plasma-SeqSensei™ SOLID CANCER IVD tool enables the identification of the maximum number of patients bearing sensitizing alterations for a tyrosine kinase inhibitor indication at diagnosis, while the custom NGS assay, with weaker clinical sensitivity, is dedicated to the exploration of resistance mechanisms and co-mutations during clinical progression.

**Abstract:**

Introduction: Progressive advanced non-small cell lung cancer (NSCLC) accounts for about 80–85% of all lung cancers. Approximately 10–50% of patients with NSCLC harbor targetable activating mutations, such as in-frame deletions in Exon 19 (Ex19del) of *EGFR*. Currently, for patients with advanced NSCLC, testing for sensitizing mutations in *EGFR* is mandatory prior to the administration of tyrosine kinase inhibitors. Patients and Methods: Plasma was collected from patients with NSCLC. We carried out targeted NGS using the Plasma-SeqSensei™ SOLID CANCER IVD kit on cfDNA (circulating free DNA). Clinical concordance for plasma detection of known oncogenic drivers was reported. In a subset of cases, validation was carried out using an orthogonal OncoBEAM^TM^ EGFR V2 assay, as well as with our custom validated NGS assay. Somatic alterations were filtered, removing somatic mutations attributable to clonal hematopoiesis for our custom validated NGS assay. Results: In the plasma samples, driver targetable mutations were studied, with a mutant allele frequency (MAF) ranging from 0.00% (negative detection) to 82.25%, using the targeted next-generation sequencing Plasma-SeqSensei™ SOLID CANCER IVD Kit. In comparison with the OncoBEAM^TM^ EGFR V2 kit, the *EGFR* concordance is 89.16% (based on the common genomic regions). The sensitivity and specificity rates based on the genomic regions (*EGFR* exons 18, 19, 20, and 21) were 84.62% and 94.67%. Furthermore, the observed clinical genomic discordances were present in 25% of the samples: 5% in those linked to the lower of coverage of the OncoBEAM^TM^ EGFR V2 kit, 7% in those induced by the sensitivity limit on the *EGFR* with the Plasma-SeqSensei™ SOLID CANCER IVD Kit, and 13% in the samples linked to the larger *KRAS*, *PIK3CA*, *BRAF* coverage of the Plasma-SeqSensei™ SOLID CANCER IVD kit. Most of these somatic alterations were cross validated in our orthogonal custom validated NGS assay, used in the routine management of patients. The concordance is 82.19% in the common genomic regions (*EGFR* exons 18, 19, 20, 21; *KRAS* exons 2, 3, 4; *BRAF* exons 11, 15; and *PIK3CA* exons 10, 21). The sensitivity and specificity rates were 89.38% and 76.12%, respectively. The 32% of genomic discordances were composed of 5% caused by the limit of coverage of the Plasma-SeqSensei™ SOLID CANCER IVD kit, 11% induced by the sensitivity limit of our custom validated NGS assay, and 16% linked to the additional oncodriver analysis, which is only covered by our custom validated NGS assay. Conclusions: The Plasma-SeqSensei™ SOLID CANCER IVD kit resulted in de novo detection of targetable oncogenic drivers and resistance alterations, with a high sensitivity and accuracy for low and high cfDNA inputs. Thus, this assay is a sensitive, robust, and accurate test.

## 1. Introduction

Lung cancers are the most common cancers [1], with a 5-year relative survival rate below 20% [2]. Among non-small cell lung cancer (NSCLC) patients, there are 10%–50% [3] of patients exhibiting epidermal growth factor receptor (EGFR), such as in-frame deletions in Exon 19 (*EGFR* Ex19del) or the *EGFR* p.L858R mutation; and Kirsten rat sarcoma viral oncogene homologue (*KRAS*) sensitizing mutations, such as the *KRAS* p.G12C. The NSCLC treatment has been modified by the availability of *EGFR* and *KRAS* predictive biomarkers [4]. Currently, for patients with advanced NSCLC, testing for sensitizing mutations is performed prior to treatment with tyrosine kinase inhibitors, such as osimertinib and sotorasib. Circulating tumor DNA (ctDNA) is isolated from blood samples [5]. In the absence of tumor tissue collection, somatic genomic analysis may be performed using ctDNA. With the increasing use of predictive biomarkers, individualized treatment is finally on the horizon.

Compared to tissue biopsies, circulating free DNA (cfDNA) is obtained with a simple and non-invasive technique causing no complications, which limits the sampling bias due to the molecular tumor burden heterogeneity [6]. Longitudinal monitoring can be performed to measure molecular disease. However, the portion of ctDNA within the total cfDNA is usually very limited, which requires a sensitive method for cfDNA analysis [7,8,9]. Most cfDNA is released from hematopoietic cells, with a short DNA length (160–180 base pairs). Currently, ctDNA detection depends on the presence of somatic alterations (deletions, insertions, amplifications, missense mutations, etc.). Although the proportion of ctDNA is often less than 1% of the total cfDNA [10], NGS technologies have been significantly improved to address this challenge, with numerous potential applications, including the detection of early molecular sensitizing alteration [9], or molecular residual disease (MRD) [11]. The clinical performance of ctDNA MRD detection has demonstrated good sensitivity for predicting disease relapse [12]. 

Several methods for cfDNA analysis enable identifying somatic alterations at high sensitivity and specificity such as digital polymerase chain reaction (ddPCR) and “BEAMing” techniques [13,14]. Next-generation sequencing (NGS) methods have also improved. Deep sequencing using tagged amplicons was successful in determining low MAF, which is a real need for the routine use of ctDNA analysis [15]. The use of flanking “bar code” sequences to uniquely identify individualized DNA fragments allows for batch processing, further reducing the analysis time.

Osimertinib, in addition to its indications in *EGFR*-mutated metastatic NSCLC, is now approved for treatments of all NSCLC with the sensitizing mutation, at all stages. In the NCT02511106 trial, osimertinib therapy has significantly increased the duration of progression-free survival (PFS) [16]. As the cfDNA concentration in the bloodstream has been suggested to be related to the tumor burden, it appears particularly important to develop extremely sensitive methods to demonstrate the presence of molecular alterations, especially for peri-operative targeted therapies [17,18].

In this study, we reported an ultra-sensitive high-throughput targeted DNA sequencing method (Plasma-SeqSensei™ SOLID CANCER IVD kit) for cfDNA identification, in terms of high clinical sensitivity detection of targeted somatic alterations. Clinical performance was assessed by comparing the Plasma OncoBEAM^TM^ EGFR V2 assay or our custom validated NGS routine assay with the Plasma-SeqSensei™ SOLID CANCER IVD assay for 212 cfDNA clinical samples collected at Lyon University Hospital (LHU, France). This work is the first study to validate the Plasma-SeqSensei™ SOLID CANCER IVD assay to determine the clinical accuracy, sensitivity, and specificity of this assay in the clinical management of NSCLC. 

## 2. Materials and Methods

### 2.1. Study Population

Patients were recruited from the clinical departments at Lyon University Hospital from 2015 to 2022. Inclusion criteria were: over 18 years old, with a histologically proven NSCLC cancer. Medical data were collected through a mandatory prescription sheet and edited by the prescribing physician. The patients were orally informed of the testing by physicians.

### 2.2. CfDNA Collection

Total blood samples were centrifuged twice, and the plasma was stored at −80 °C until cfDNA extraction and molecular analyses [19]. CfDNA was extracted using the QIA-amp Circulating Nucleic Acid kit (Qiagen, Valencia, CA, USA, Cat No 55114). CfDNA were quantified using a Qubit™ 4 Fluorometer (Invitrogen™, Cat No Q33238, Carlsbad CA, 92008, USA) with the Qubit™ dsDNA HS Assay kit (Invitrogen™, Cat No 32854). 

### 2.3. Library Preparation for DNA Sequencing

For our custom validated NGS library preparation, 10–100 ng cfDNA were used (custom capture technology from SOPHiA GENETICS, Lausanne, Switzerland), according to the manufacturer’s instructions (SOPHiA GENETICS) [20]. The custom panel covered 77 genes (updated from our 66 gene V1 panel used in Garcia et al., 2021) in routine use in our medical laboratory. The libraries were sequenced on NextSeq 550 (Illumina technology, San Diego, CA 92122, USA) in 2 × 150 paired-end runs. The bioinformatics analysis was performed using the SOPHiADDM platform, according to the validation method described in Bieler J, 2021 [21] and based on the associated guidelines [22].

### 2.4. OncoBEAM^TM^ EGFR V2 Kit for DNA Exploration

CfDNA was analyzed using the OncoBEAM^TM^ EGFR V2 kit (Sysmex Inostics, Hamburg, Germany, Cat No ZR150220), according to the manufacturer’s instructions. The OncoBEAM^TM^ EGFR V2 kit is able to detect 36 alterations, including p.G719X, p.L858R, and p.L861Q alterations, as well as 27 of the deletions in Exon 19, p.T790M, and p.C797S *EGFR* resistance mutations. The analysis was performed using Sysmex Inostics software (FCS Express 5 Flow v.5.01.0082) dedicated to BEAMing technology [20].

### 2.5. Targeted Next-Generation Sequencing Plasma-SeqSensei™ SOLID CANCER IVD Kit for DNA Sequencing

CfDNA libraries were generated using the Plasma-SeqSensei™ SOLID CANCER IVD kit (Sysmex Inostics GmbH, Cat. No ZR150510). For this assay, the process was partially carried out using an EpMotion 5075t NGS robot from Eppendorf (Hamburg, Germany), as well as the Applied Biosystems™ Veriti™ Dx 96-well Thermal Cycler (Foster City, CA, USA). Internally, the analytical performance (using Horizon’s commercial controls) was validated, according to the associated guidelines for the validation of NGS, based the oncology panels [22]. 

The Plasma-SeqSensei™ SOLID CANCER IVD assay covers the *BRAF* alterations in the partial exons 11 and 15 (amino acid residues: 462–477, 582–604), the *EGFR* alterations in the partial exons 18, 19, 20, and 21 (amino acid residues: 706–725, 743–759, 762–775, 788–801, 856–87), the *KRAS* alterations in the partial exons 2, 3, and 4 (amino acid residues: 12–34, 57–76, 110–117, 141–148), and the *PIK3CA* alterations in the partial exons 10 and 21 (amino acid residues: 538–553, 1040–1065). Identification of the somatic alterations was performed using Sysmex Inostics software (Plasma SeqSensei IVD Software v1.1.7) and the Sysmex Inostics’ bioinformatics pipeline.

### 2.6. Statistics

Descriptive statistics were obtained using the GraphPad InStat software, version 9.4.1 (La Jolla, CA, USA).

## 3. Results

### 3.1. Description of the Patient Cohort

To compare clinical performances of the assays, we performed molecular profiling on paired plasma samples. We did not include clinical data in this study testing clinical performance. The cfDNA molecular profiles are reported in Table 1. 

We analyzed 215 somatic alterations identified by at least one of the three assays in 123 plasma samples. Additionally, 89 samples showed no somatic alterations in the limited panel coverages of the three assays. There were some samples with co-alterations (two alterations or more per sample, see Appendix A). Lacking residual biological materials, 82 samples were excluded from the Plasma OncoBEAM^TM^ EGFR V2 analysis. A total of 130 samples were analyzed with the Plasma OncoBEAM^TM^ EGFR V2 assay, and 209 samples were analyzed with our custom validated NGS routine assay. All samples (n = 212) were analyzed using the Plasma-SeqSensei™ SOLID CANCER IVD kit.

As shown in Table 1, the proportion of somatic alterations in the cohort was reported for the principal oncodrivers in at least one assay performed in this study. The choice of the samples was based on the known molecular profiles obtained with the routine Plasma OncoBEAM^TM^ EGFR V2 or our custom validated NGS assays. We aimed to obtain a near-equal number (n = 123, clinical sensitivity analysis) of mutated and not mutated samples (n = 89, clinical specificity analysis) in each group to facilitate the clinical performance analysis. Therefore, this is not representative of the clinical proportions of alterations found in NSCLC cancers. 

The full description of all somatic alterations is reported in Appendix A. The specificity was addressed with the wild-type sample status, determined using two orthogonal reference methods (Plasma OncoBEAM^TM^ EGFR V2 and our custom validated NGS assay), and reported in Appendix A.

We chose samples with various *EGFR* alterations that had been useful in patient management. In particular, we included resistance mutations that usually occur in low abundance and lead to a therapeutic switch. A total of 23 samples were chosen for their low MAF (below 1%) from the positive cases using the Plasma OncoBEAM^TM^ EGFR V2 reference technique in the field (Appendix A).

### 3.2. Description of the cfDNA Input 

In Table 2, we summarize the median and mean values of cfDNA inputs in the three assays, in their clinical use conditions (the detailed data are fully reported in Appendix A). The number of tests per sample is different for some samples, due to insufficient available material. Individual results are reported in Appendix A. The median cfDNA inputs observed in our custom validated NGS routine assay is usually higher than in the other methods, as we compensated for the lack of sensitivity by increasing the cfDNA input in the test. This condition is included in routine testing, and only one discording sample was analyzed using low cfDNA input in our custom validated NGS routine, compared to the Plasma-SeqSensei™ SOLID CANCER IVD kit. Our custom validated NGS routine’s decreased sensitivity cannot be attributed to the cfDNA inputs (Appendix A). For the Plasma-SeqSensei™ SOLID CANCER IVD kit, we were technically limited by the need to standardize the number of samples per run for the Illumina’s NextSeq500 sequencing step for its clinical use. This is fully representative of the feasible workflow in routine use. We defined the maximal quantity of cfDNA at 86 ng/sample (limit for the flow cell capability and the technique’s recommendations) in the Plasma-SeqSensei™ SOLID CANCER IVD kit. Nevertheless, the cfDNA input of most of samples was usually around 36 ng. For the Plasma OncoBEAM^TM^ EGFR V2, the range of feasible cfDNA was larger: between 1 ng to 10,000 ng. 

### 3.3. Comparison of the Three Assays Based on the Somatic Alterations Found

Taking the Plasma-SeqSensei™ SOLID CANCER IVD kit as reference and considering only the genetic alterations covered by the Plasma OncoBEAM^TM^ EGFR V2, the sensitivity and specificity are, respectively, 0.84 and 0.94 (Table 3). Similarly, considering only the genetic alterations covered by the Plasma-SeqSensei™ SOLID CANCER IVD kit, the sensitivity and specificity are, respectively, 0.83 and 0.76 (Table 3).

We described the cases of discordances from the three assays, according to the coverage criteria and the sensitivity level of each test. In Figure 1, we show a global rate (over all genes studied) of 75% and 68% of clinical concordance between the Plasma-SeqSensei™ SOLID CANCER IVD assay with the Plasma OncoBEAM^TM^ EGFR V2 assay and our custom validated NGS assays. 

In comparison with the OncoBEAM^TM^ EGFR V2 kit, in Figure 1A, we observed a 75% clinical concordance with the Plasma-SeqSensei™ Solid Cancer IVD kit. The genomic discordances were observed in 25% of the samples: 5% in those linked to the limit of coverage of the OncoBEAM^TM^ EGFR V2 kit; 7% in those linked to the limit of *EGFR* sensitivity of the Plasma-SeqSensei™ SOLID CANCER IVD Kit (MAF = 0.14% for sample ID1594 and MAF = 0.15% for sample ID1519 determined with the OncoBEAM^TM^ EGFR V2 kit, from Appendix A); and for 13% in the samples linked to the larger *KRAS*, *PIK3CA*, and *BRAF* coverage of the Plasma-SeqSensei™ SOLID CANCER IVD kit. Benefitting from the higher coverage from targetable *BRAF*, *KRAS*, and *PIK3CA* targetable alterations, the plasma targeted NGS Plasma-SeqSensei™ Solid Cancer IVD kit showed a 13% discordance with the Plasma OncoBEAM^TM^ EGFR V2 kit. Most of these somatic alterations were cross validated in our orthogonal custom validated NGS assay used in the routine management of patients.

In comparison with our custom validated NGS assay, described in Figure 1B, we observed a 68% clinical concordance with the Plasma-SeqSensei™ Solid Cancer IVD kit for all cfDNA inputs. The genomic discordances were observed in 32% of the samples: 5% due to the limit of coverage of the Plasma-SeqSensei™ Solid Cancer IVD kit; 11% due to the limit of sensitivity of our custom validated NGS assay; and 16% due to the additional oncodriver analysis, which is only covered by our custom validated NGS assay. Our custom validated NGS assay sensitivity is determined (according the methodology described into Bieler et al.) [21] for an MAF at 1% at 10–20 ng of cfDNA input and at 0.4% for 20–150 ng cfDNA inputs, explaining the limit of sensitivity.

We show the MAF clinical correlation in the three assays (Figure 2). The r squared simple linear regression of the Plasma-SeqSensei™ SOLID CANCER IVD kit was respectively calculated at 0.46 and 0.74 with our custom validated NGS assay (*p* < 0.0001) and the Plasma OncoBEAM^TM^ EGFR V2 kit (*p* < 0.0001). 

A full description of the clinical discordances between the Plasma OncoBEAM^TM^ EGFR V2 kit and the Plasma-SeqSensei™ SOLID CANCER IVD kit are reported in Appendix A. The discordances between the Negative Plasma-SeqSensei™ SOLID CANCER IVD kit and the Positive Plasma OncoBEAM^TM^ EGFR V2 kit for EGFR targeted alterations covered by the Plasma OncoBEAM^TM^ EGFR V2 kit were due to an MAF lower than 0.15% (Appendix A). In these cases, for the technical limitations described in the Plasma-SeqSensei™ SOLID CANCER IVD kit instructions, or due to limits in the quantity of available residual biological material, we could not always analyze the samples with identical cfDNA inputs. In these cases, the lower input of cfDNA used in the Plasma-SeqSensei™ SOLID CANCER IVD assay (in Appendix A, the reported ratio of cfDNA input is less than one) may contribute to the absence of detection, in comparison to Plasma OncoBEAM^TM^ EGFR V2 technology. Most of the discordance in the Plasma-SeqSensei™ SOLID CANCER IVD kit was observed in the context of MRD (indicated as P for “during progression” in Appendix A). The other discordances with the Plasma OncoBEAM^TM^ EGFR V2 assay occurred in other genomic regions in *EGFR*, *KRAS*, *PIK3CA*, and *BRAF*. These were predicted and were due to the lower coverage of this digital PCR method compared to the Plasma-SeqSensei™ SOLID CANCER IVD kit. For most of the discordant samples, the cfDNA inputs were in the same range in both assays. We included 82 additional samples that were only analyzed using our custom validated NGS assay and the Plasma-SeqSensei™ SOLID CANCER IVD assay, containing the alterations for *KRAS*, *BRAF*, and *PIK3CA (*Appendix A). No data was reported for the Plasma OncoBEAM^TM^ EGFR V2 assay in these samples, since these genes are not covered by the assay. We analyzed 209 samples in this comparison, as reported in Appendix A. These discordances were found in the covered genes using the Plasma-SeqSensei™ SOLID CANCER IVD assay. We reported that most of the discordances of our custom validated NGS assay were observed with an MAF percentage lower than 1% (Appendix A), due to the lower sensitivity of our custom validated NGS assay. The negative results reported by the Plasma-SeqSensei™ SOLID CANCER IVD assay were due to the lower coverage of the panel, and were not discovered in regions bearing targetable alterations. A false negative case (Sample ID 2441) was reported, due to the very low cfDNA input (3.44 ng in Plasma-SeqSensei™ SOLID CANCER IVD assay, 2-fold lower input compared to our custom validated NGS assay) (Appendix A). The second false negative case was sample ID2387, with the no detection of *BRAF* p.D594N found at a 1.5% MAF using our custom validated NGS assay. This may a false positive detected with our custom validated NGS assay. For most of the discordance samples, the cfDNA inputs were in the same range in both assays (Appendix A).

Finally, we evaluated the alterations found by our custom validated NGS assay in genes not included in the Plasma-SeqSensei™ SOLID CANCER IVD panel (Appendix A). The MAF is highly variable, and detailed alterations are mainly in the *TP53* gene, often in co-mutations (Appendix A). Some alterations in the *ERBB2* and *MET* genes are associated with prognostic and/or predictive values.

## 4. Discussion 

*EGFR* assays on cfDNA biopsies routinely allow physicians to detect targetable or prognostic somatic alterations at early stages such as in peri-operative contexts, and in residual disease for patients with NSCLC following treatment with curative intent [23]. This can be achieved by monitoring testing. Molecular cfDNA testing is encouraged where sequential tissue biopsies and tumor tissue testing may cause undue harm [24]. Here, our data strongly supports that a highly sensitive and specific tool, such as the Plasma-SeqSensei™ SOLID CANCER IVD kit, can identify more patients bearing targeted alterations in their tumoral genome at diagnosis for tyrosine kinase inhibitor indication in the first intention. Conversely, the custom NGS assay, with its weaker sensitivity, is better adapted to the exploration of resistance mechanisms and co-mutations. One limitation of the Plasma-SeqSensei™ SOLID CANCER IVD kit is the non-detection of insertion into the *ERBB2* gene, which is addressed by some drugs currently in clinical trials. 

This is supported by the clinical need for identifying patients with *EGFR* sensitizing alterations to benefit from osimertinib in early NSCLC stages (ADAURA, NCT02511106). Accordingly, it is important to use ultra-sensitive molecular NGS assays, which do not sacrifice specificity, for patients with non-contributive tissue molecular analysis. Tissue sample analysis is mandatory for verifying cancer types identified only via histological analysis, determining to the choice of clinical management and treatments. Therefore, although molecular tissue analysis remains the reference for cancer diagnosis, the rapid selection required for administering targeted therapies can be better achieved via cfDNA testing. A cfDNA positivity of 24%, 77%, and 87% has recently been shown before treatment in patients with stage I, II, and III disease, respectively [9]. Interestingly, cfDNA detection showed a clinical specificity of >98.5% before clinical detection of recurrence of the primary tumor. CfDNA was still detected after treatments in most patients. These molecular recurrences, detected in the cfDNA, are a complement to tumor tissue biopsy analysis and can identify alterations of resistance, such as *EGFR* p.T790M and *EGFR* p.C797S. This is especially important for the prescription of tyrosine kinase inhibitors, where management decisions for patients with NSCLC and their treatment can be modified within 10 days. In our report, we showed that the Plasma-SeqSensei™ SOLID CANCER IVD assay reported all *EGFR* targeted alterations with a clinical MAF superior to 0.15% (determined with the BEAMing technique), according the cfDNA input (10–81 ng). The benefit of a higher MAF sensitivity for targetable genomic alterations could be significant, especially for early-stage lung cancers and MRD monitoring [11,25,26]. This was supported by the fact that, for most solid tumors, cfDNA levels are limited, with a MAF less than 5% in advanced cancers, and less than 1% in early stages [27]. CfDNA levels decrease further after curative-intent treatment, when MAF levels are often low. To measure low MAF determination (below 1%), a sequencing system (Safe-SeqS, Hambourg, Germany) was developed based on a PCR amplification of selected genomic regions prior to NGS analysis. Tagged-amplicon sequencing (TAm-Seq) is shown to detect somatic alterations, with a MAF lower than 2%. To improve cfDNA analysis, the integration of a unique molecular identifier (UMI)-based multiplex PCR, followed by NGS, called Safe-SeqS, was set up. Each DNA fragment is tagged with a UMI, allowing for the elimination of PCR technical errors [15]. This approach enabled the highly sensitive and specific detection of rare somatic alterations. Safe-SeqS was validated in clinical cohorts [28]. In the field, based on a similar methodology, we evaluated the Plasma-SeqSensei™ SOLID CANCER IVD assays. The reported clinical accuracy was 100% for an MAF above 0.15%, with high specificity (determined in those who were driver-negative via the Plasma OncoBEAM^TM^ EGFR V2 kit). This new method has a larger coverage of targeted somatic alterations in NSCLC.

In the field, we previously integrated the Plasma OncoBEAM^TM^ EGFR V2 kit into routine clinical use to achieve this high sensitivity, with high accuracy [23]. We also optimized the bioinformatics analysis of our custom validated NGS assay, based on the capture approach [29]. In this study, we highlighted the high sensitivity and specificity of the Plasma-SeqSensei™ SOLID CANCER IVD assays for cfDNA analysis, particularly their higher performance than our custom validated NGS assay for the detection of targetable *EGFR* alterations. The limited coverage of the Plasma-SeqSensei™ SOLID CANCER IVD assay for some targeted genes explained the remaining portion of the discordances observed between the Plasma-SeqSensei™ SOLID CANCER IVD and our custom validated NGS assays. As these discordances concerned mainly non-targetable genomic alterations, the impact on clinical practice could be limited. The application of our large custom validated NGS assay allows clinicians to search for prognostic biomarkers or monitor MRD in the absence of targetable somatic alterations.

Somatic alterations need to be cross-differentiated from the clonal hematopoiesis of indeterminate potential (CHIP), which is often present in elderly healthy persons due to the accumulation of genetic modifications in hematopoietic cells [30,31]. In measuring cfDNA, false-positive findings may be due to the detection of non-reference variants. The rate of false positives is strongly increased for a cfDNA MAF lower than 1%, especially in MRD monitoring. The majority of CHIP variants mainly occur in *TP53*, which is challenging, given its high prevalence as a driver mutation in solid tumors, and is commonly tracked in cfDNA [32]. In accord with this observation, *TP53* non sensitizing alterations in our cfDNA study could be part of CHIP (Appendix A).

## 5. Conclusions

Our continued commitment to using an effective focal plasma-targeted next-generation sequencing assay (Plasma-SeqSensei™ SOLID CANCER IVD testing kit) for the rapid molecular testing of predictive markers is still needed in routine clinical practice for the early stages of NSCLC, with a targeted therapy indication. We highlighted a satisfactory clinical accuracy of the kit towards low cfDNA inputs. This assay improves screening or diagnostic testing in the early and curable stages of NSCLC [33]. The limit can be overcome by the parallel use of our custom validated NGS assay to cover larger genomic regions.

## Figures and Tables

**Figure 1 cancers-15-01574-f001:**
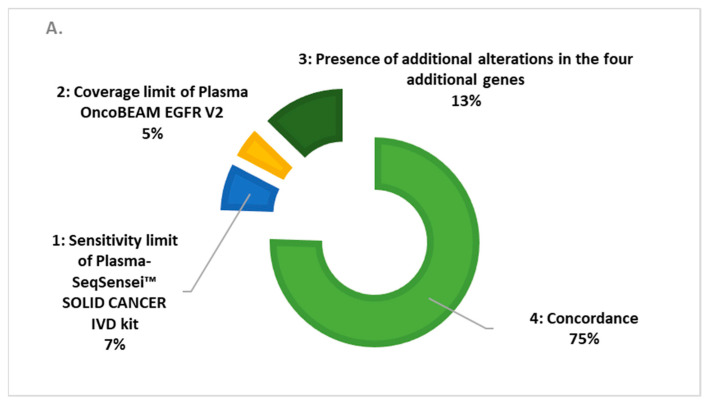
Cases of discordance (%) in the three assays: (**A**) Comparison of the Plasma OncoBEAM™ EGFR V2 assay vs. the Plasma-SeqSensei™ SOLID CANCER IVD assay; (**B**) comparison of our custom validated NGS assay vs. the Plasma-SeqSensei™ SOLID CANCER IVD assay.

**Figure 2 cancers-15-01574-f002:**
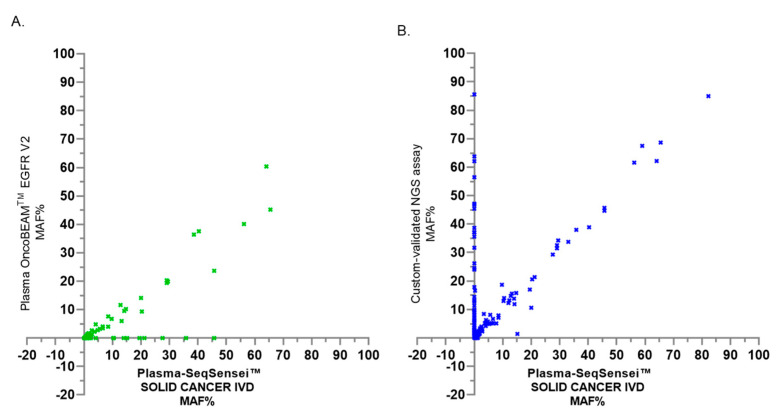
MAF clinical correlation of the Plasma OncoBEAM^TM^ EGFR V2 assay (**A**) and our custom validated NGS assay (**B**) vs. the Plasma-SeqSensei™ SOLID CANCER IVD assay. The MAF is between 0% (no detection of somatic alterations) to 85% (positive detection of alterations using one of the three assays).

**Table 1 cancers-15-01574-t001:** cfDNA characteristics of the population and percentages (%) of the mutation per gene found in the cohort.

Genes	Somatic Alterations	%
Wild-Type (WT) for all assays (no detected alterations) in the sample (n = 89)		
*EGFR*	116	53.85
*KRAS*	25	11.63
*PIK3CA*	14	6.51
*BRAF*	10	4.65
Other genes	50	23.26
	**Total = 215**	

**Table 2 cancers-15-01574-t002:** CfDNA input of the population for the three assays.

	Plasma OncoBEAM^TM^ EGFR V2	Plasma-SeqSensei™ SOLID CANCER IVD Kit	Our Custom Validated NGS Assay
	**DNA Content**	**DNA Content**	**DNA Content**
	**[ng Input]**	**[ng Input]**	**[ng Input]**
Number of analyzed samples	130.00	212.00	209.00
Minimum	2.66	2.10	8.49
25% Percentile	11.96	17.00	25.10
**Median**	**22.40**	**33.83**	**41.1**
75% Percentile	46.87	36.81	65.90
Maximum	5165.00	85.00	150.00
Range	5162.00	82.90	141.50
**Mean**	**81.84**	**30.56**	**51.34**
Std. Deviation	453.10	18.23	37.46
Std. Error of Mean	39.74	1.252	2.59

**Table 3 cancers-15-01574-t003:** Performance of the assays. A. Focus on *EGFR* targeted clinical alterations covered by the Plasma OncoBEAM^TM^ EGFR V2, given by Plasma OncoBEAM^TM^ EGFR V2 and Plasma-SeqSensei™ SOLID CANCER IVD assays; B. focus on *KRAS*, *EGFR*, and *PIK3CA* genes covered by the Plasma-SeqSensei™ SOLID CANCER IVD kit vs. our custom validated NGS assays. Sensitivity and specificity calculation based on https://probabilitycalculator.guru/sensitivity-and-specificity-calculator/ (accessed on 29 December 2022). PPA and PPV calculation according to Jennings, 2017 [22].

A. **Targeted and Covered Alterations by Both Compared Assays**	Plasma OncoBEAM^TM^ EGFR V2 NEGATIVE Status	Plasma OncoBEAM^TM^ EGFR V2 POSITIVE Status	
**Plasma-SeqSensei™ SOLID CANCER IVD Kit NEGATIVE status**	72	14	**Sensitivity = 0.84**
**Plasma-SeqSensei™ SOLID CANCER IVD Kit POSITIVE status**	4	77	**Specificity = 0.94**
Positive likelihood ratio = Sensitivity/(1 − Specificity) = 16.087 Negative likelihood ratio = (1 − Sensitivity)/Specificity = 0.044 Positive percentage agreement [PPA = 77/(77 + 14) = 84.6%] Positive predictive value [PPV = 77/(7 7+ 4) = 95.06%]			
**B.** **Genes and Alterations Covered by Both Compared Assays**	**Our Custom Validated NGS Assay** **NEGATIVE Status**	**Our Custom Validated NGS Assay POSITIVE Status**	
**Plasma-SeqSensei™ SOLID CANCER IVD Kit NEGATIVE status**	102	21	**Sensitivity = 0.83**
**Plasma-SeqSensei™ SOLID CANCER IVD Kit POSITIVE status**	32	101	**Specificity = 0.76**
Positive likelihood ratio = Sensitivity/(1 − Specificity) =3.466 Negative likelihood ratio = (1 − Sensitivity)/Specificity = 0.131 Positive percentage agreement [PPA = 101/(101 + 21) = 82.7%] Positive predictive value [PPV = 101/(101 + 32) = 75.9%]			

## Data Availability

The data presented in this study are available in the Appendix A and from the corresponding author: lea.payen-gay@chu-lyon.fr.

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
