# Peer review of "Paired Comparison of Routine Molecular Screening of Patient Samples with Advanced Non-Small Cell Lung Cancer in Circulating Cell-Free DNA Using Three Targeted Assays"

_cancers, 2023, doi:10.3390/cancers15051574_

Round 1

Reviewer 1 Report

This research compares three platforms for targeted sequencing of circulating tumour DNA designed for known lung cancer genes. It would be useful to lung cancer researchers to know which short variants are missed by which platform. The issues I have identified are:

• The article title is too long and hard to read. Instead of ".. Using OncoBEAMTM EGFR V2, Targeted Next‐Generation Sequencing Plasma-SeqSensei™ SOLID CANCER IVD Kit and custom-validated NGS Assay" this could be shortened to "... Using Three Targeted Assays".
• "The bioinformatics analysis was described in technical part de-scribing each technics." isn't clear, at least to a native English speaker such as me.
• "... while the more blue the color intensity is the more important the MAF is." I don't like the use of the word "important" because it is subjective. Just use "higher" instead of it.
• Table 3 and Table 4 are more like raw data that a summarised result and are very large. Perhaps they should not be in the main text but supplementary tables instead. The ID Samples column has no meaning to a reader who was not involved with the project, for example. Table 5 is a concise summary and what I would expect to see in the main text.
• Data Availability Statement is "The data presented in this study are available on request from the corresponding author." However, this is problematic because academic researchers often change jobs and universities in which they work, so the e-mail address contained in the journal article is closed when the researcher stops working for that particular university. Please write a more appropriate data availability statement.
• Conflict of Interest Statement is "The authors declare no conflict of interest.". However, "We thank AstraZeneca and Sysmex Inostics for financial support" and two of the assays used in the benchmarking is developed by Sysmex Inostics:
1. "CfDNA libraries were generated using the Plasma-SeqSensei™ SOLID CANCER IVD Kit (Sysmex Inostics GmbH, Cat. No ZR150510)."
2. "CfDNA was analyzed using the OncoBEAMTM EGFR V2 kit (Sysmex Inostics, Ham-burg, Germany, Cat No ZR150220)"
So, I think there is a concerning conflict of interest with this study.

Author Response

Reviewer 1 :

This research compares three platforms for targeted sequencing of circulating tumour DNA designed for known lung cancer genes. It would be useful to lung cancer researchers to know which short variants are missed by which platform. The issues I have identified are:
• The article title is too long and hard to read. Instead of ".. Using OncoBEAMTM EGFR V2, Targeted Next‐Generation Sequencing Plasma-SeqSensei™ SOLID CANCER IVD Kit and custom-validated NGS Assay" this could be shortened to "... Using Three Targeted Assays".

Answer. We agreed with this comment. We shortened the title as recommended

  • "The bioinformatics analysis was described in technical part de-scribing each technics." isn't clear, at least to a native English speaker such as me.

Answer : We clarified  this section and completed the material and method section

  • "... while the more blue the color intensity is the more important the MAF is." I don't like the use of the word "important" because it is subjective. Just use "higher" instead of it.

Answer. We agreed with this comment. We made the correction.

  • Table 3 and Table 4 are more like raw data that a summarised result and are very large. Perhaps they should not be in the main text but supplementary tables instead. The ID Samples column has no meaning to a reader who was not involved with the project, for example. Table 5 is a concise summary and what I would expect to see in the main text.

Answer. We agreed with this comment. We transferred table 3 and table 4 into the supplementary data section. It is now Table S1 and Table S2.

  • Data Availability Statement is "The data presented in this study are available on request from the corresponding author." However, this is problematic because academic researchers often change jobs and universities in which they work, so the e-mail address contained in the journal article is closed when the researcher stops working for that particular university. Please write a more appropriate data availability statement.

Answer. We agreed with this comment. All the data are available in the supplementary data.

  • Conflict of Interest Statement is "The authors declare no conflict of interest.". However, "We thank AstraZeneca and Sysmex Inostics for financial support" and two of the assays used in the benchmarking is developed by Sysmex Inostics:
    1. "CfDNA libraries were generated using the Plasma-SeqSensei™ SOLID CANCER IVD Kit (Sysmex Inostics GmbH, Cat. No ZR150510)."
    2. "CfDNA was analyzed using the OncoBEAMTM EGFR V2 kit (Sysmex Inostics, Ham-burg, Germany, Cat No ZR150220)"
    So, I think there is a concerning conflict of interest with this study.

Answer. We agreed with this comment. We changed the statement.

Reviewer 2 Report

Barthelemy David et all are describing a comparison of 3 different clinical cfDNA assays for NSCLC molecular screening. Although the work is interesting, several conceptual errors have been made in the data analysis. The validation of a clinical test follows specific guidelines that help ensure a standardization level among clinical assays. I made several comments on the pdf file to help the authors but i strongly suggest the authors to read some of those guidelines before revised the paper for re-submission. Here some of the references:

1. Guidelines for Validation of Next-Generation Sequencing–Based Oncology Panels 10.1016/j.jmoldx.2017.01.011

2. Recommendations for the use of next-generation sequencing (NGS) for patients with metastatic cancers: a report from the ESMO Precision Medicine Working Group. https://doi.org/10.1016/j.annonc.2020.07.014

Author Response

Barthelemy David et all are describing a comparison of 3 different clinical cfDNA assays for NSCLC molecular screening. Although the work is interesting, several conceptual errors have been made in the data analysis. The validation of a clinical test follows specific guidelines that help ensure a standardization level among clinical assays. I made several comments on the pdf file to help the authors but i strongly suggest the authors to read some of those guidelines before revised the paper for re-submission.

 Here some of the references:

  1. Guidelines for Validation of Next-Generation Sequencing–Based Oncology Panels 10.1016/j.jmoldx.2017.01.011
  2. Recommendations for the use of next-generation sequencing (NGS) for patients with metastatic cancers: a report from the ESMO Precision Medicine Working Group. https://doi.org/10.1016/j.annonc.2020.07.014

Answer : we made the modifications and corrections suggested by the reviewer (in the reviewer’s pdf file), including the two references suggested, and made a native proofreading of the revised manuscript. We highlighted in blue de principal changes in the manuscript. We added the references of the three assays into the material section. We performed a clinical performance evaluation in this manuscript. The analytical performances had been previously published or done by the provider (and just checked internally before using the assay in routine.

Round 2

Reviewer 1 Report

The authors have addressed my issues.